# Dynamic Wildfire Navigation System

**Mitsuhiro Ozaki [1],*, Jagannath Aryal [1] and Paul Fox-Hughes [2]**

[1]  Geography and Spatial Sciences, University of Tasmania, Private Bag 76, Hobart, Tasmania 7001, Australia; jagannath.aryal@utas.edu.au

[2]  Bureau of Meteorology, GPO Box 727, Hobart, Tasmania 7000, Australia; paul.fox-hughes@bom.gov.au

*  Correspondence: mozaki@utas.edu.au

**Abstract:** Wildfire, a natural part of many ecosystems, has also resulted in significant disasters impacting ecology and human life in Australia. This study proposes a prototype of fire propagation prediction as an extension of preceding research; this system is called "Cloud computing based bushfire prediction", the computational performance of which is expected to be about twice that of the traditional client-server (CS) model. As the first step in the modelling approach, this prototype focuses on the prediction of fire propagation. The direction of fire is limited in regular grid approaches, such as cellular automata, due to the shape of the uniformed grid, while irregular grids are freed from this constraint. In this prototype, fire propagation is computed from a centroid regardless of grid shape to remove the above constraint. Additionally, the prototype employs existing fire indices, including the Grassland Fire Danger Index (GFDI), Forest Fire Danger Index (FFDI) and Button Grass Moorland Fire Index (BGML). A number of parameters, such as Digital Elevation Model (DEM) and forecast weather data, are prepared for use in the calculation of the indices above. The fire study area is located around Lake Mackenzie in the central north of Tasmania where a fire burnt approximately 247.11 km$^2$ in January 2016. The prototype produces nine different prediction results with three polygon configurations, including Delaunay Triangulation, Square and Voronoi, using three different resolutions: fine, medium and coarse. The Delaunay Triangulation, which has the greatest number of adjacent grids among three shapes of polygon, shows the shortest elapsed time for spread of fire compared to other shapes. The medium grid performs the best trade-off between cost and time among the three grain sizes of prediction polygons, and the coarse size shows the best cost-effectiveness. A staging approach where coarse size prediction is released initially, followed by a medium size one, can be a pragmatic solution for the purpose of providing timely evacuation guidance.

**Keywords:** GIS; FDI; wildfire; PostGIS; GeoDjango

## 1. Introduction

This study has three aims: (1) it identifies limitations in the development of fire prediction models using cellular automata; (2) it suggests a solution to these limitations and implements an example prototype as a preliminary approach, concentrating on the functionality of fire propagation prediction in detail using geographical information system (GIS) software; and (3) it proposes an efficient strategy of evacuation guidance from wildfires. In particular, the last aim is important because there is a trade-off between quality of prediction and the performance speed of fire calculation. For instance, a high-quality simulation would take too long to execute to be useful for urgent alerting. On the other hand, poor-quality guidance should not be released, even if the prediction is calculated rapily. To address this problem, various predictions with different geometries and sizes of polygons are developed in this study by employing various Fire Danger Indices (FDIs) and ingesting parameters, including the Digital Elevation Model (DEM), forecast weather data and vegetation, storing the

predictions in a database. The predictions will then be verified on an example Tasmanian wildfire to establish the best cost-effectiveness and timeliness among these predictive polygons. At first, the study area and prototype architecture are illustrated, then the methodologies, such as calculation of FDIs, prediction of fire propagation, and verification are addressed. Prediction results are then analyzed, and the impacts of grid sizes and shape, as well as spatial data, are discussed, leading to recommendations for implementation. Lastly the limitations of this study and possibilities for future work are mentioned.

Globally, wildfires have devastated lives and properties. For instance, a state-wide, fire cost the lives of 44 people and devastated about 100,000 hectares in California, USA in 2017 [1]. In Portugal in the same year, there were more than 62 casualties in the forest fire of Pedrógão Grande area [2]. In Australia, a death toll of 173 resulted from the Black Saturday wildfires in Victoria in 2009 [3]. In 1983, 75 casualties were caused by Ash Wednesday in Victoria and South Australia [4]. The Black Tuesday wildfires in 1967 cost 62 human lives in Southern Tasmania [5]. To mitigate the casualty toll from wildfires, a dynamic guidance system called "Cloud computing based bushfire prediction" has been proposed to assist people in evacuating from approaching wildfires [6]. This system will be based on a cloud computing platform so that the expected computational performance will be about twice that of the traditional client-server (CS) model, by employing concurrent processing with multiple computer units [6]. Note that the focus in this work is on fire propagation only, while other features, such as parallel calculation using cloud computing, are out of scope.

A number of research studies have been conducted on fire spread simulations. A recent study addresses the improvement of accuracy by re-categorization of fuel into five types: grasslands, temperate shrublands, semi-arid shrublands, dry eucalypt forests and conifer forests [7]. This approach has been employed in developing the prototype in this study. In fact, this prototype is capable of configuring the fuel load and risk of ignition (flammability) by vegetation groups. Some fire behavior models distinguish FDIs between forest and grassland. For example, Phoenix employs both the Forest Fire Danger Index (FFDI) and Commonwealth Scientific and Industrial Research Organisation (CSIRO) southern grassland fire spread models [8]. This prototype, however, employs the Button Grass Moorland Index (BGML), in addition to the McArthur Grassland Fire Danger Index (GFDI) and Forest Fire Danger Index (FFDI). This is because buttongrass is highly flammable and widespread in Tasmania, which necessitates predicting its fire spread separately from other vegetation types [9,10]. Some models employ geo-database software for scalability given the large spatial datasets used. In addition, fire spread model output can be viewed and shared through a web browser. For example, the WIFIRE project is implemented with PostGIS, which is an extension of the relational database management system, and a web platform called Firemap [11]. The architecture from that project [11] is adapted in this prototype. In addition, GeoDjango, the spatial web application framework powered by Django [12], calculates fire spread so that the results can be viewed through a browser in future implementations. Cellular automata are used in some models for the core fire spread modelling functionality, noting their high efficiency in the calculation of fire spread [13], and their further capability of modelling fire spread under heterogeneous conditions, such as varying wind direction [14]. However, cellular automata cause a distortion of fire spread because of their uniform shape, usually square or hexagonal grids. For instance, the direction of fire is constrained into either of eight directions only in square grids. However, this distortion of fire orientation can be removed by the use of an irregular grid [15]. In this prototype, the grid-based approach is adapted from the above studies [15]. In this implementation, fire propagation is simulated not based on the shape of the grid but on its centroid, so that the fire orientation is freed from dependence on shape. To verify the independence of the grid shape, both regular and irregular grids are generated in various scales. Further, grid data such as travel time and fire status are stored not in Random Access Memory (RAM) but in the database, so that the simulation can stop and restart as required. Additionally, topographically dynamic wind is an important factor for fire management because most atmospheric predictions are at a sufficiently high resolution for the prediction of fire behavior, and even a weak fire is prone to be influenced by terrain-affected

winds [16,17]. Therefore, this prototype employs WindNinja, which generates a topographically informed grid for both wind direction and wind magnitude, so that fire behavior models can digest topographically dynamic wind instead of raw wind forecast data when predicting fire spread [18]. There are some prominent integrated fire prediction systems. For example, CSRIO Spark has been developed with OpenCL so that the system can run in multiple computational environments based on empirical approaches [19,20]. In addition, this system can be executed rapidly because it is written in C language, which is a compiled to machine language with CPU or GPU optimization [20,21].

The recommended approach with the best grid size and shape from the discussion in this prototype will be available to apply in future implementations of "Cloud computing-based bushfire prediction" to facilitate evacuation from wildfires.

## 2. Study Area and System Architecture

This section describes the study area and system specification and architecture. The details of the data structure are described in the section: Data Structure, in the Supplementary file.

### 2.1. Study Area

As a wildfire example, the January 2016 fires in Tasmania are examined. In particular, the meteorological grid data between 19th and 22nd January 2016, and topographic data for the fire around Lake Mackenzie, identified as Lake Mackenzie Road Fire in the database, are ingested to simulate the wildfire [22]. However, these source data not only cover this period but extend until 3rd February 2016 as a temporal buffer to simulate several periods of fire activity until the predicted fire reaches at least the size of the actual fire event, in various grid configurations. The total area of this wildfire was approximately 247.11 km$^2$, as assessed on 4th May 2016 (Figure 1).

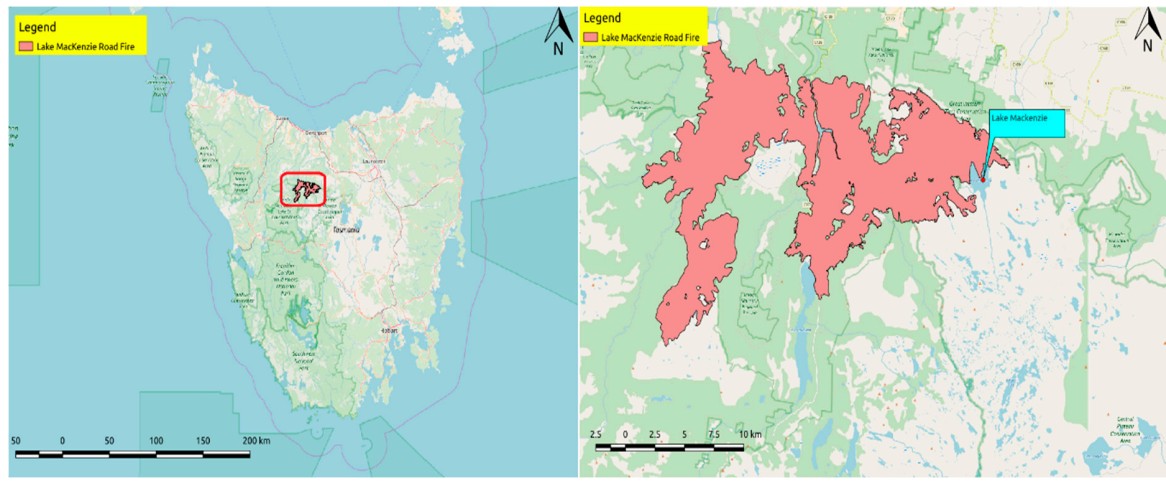

**Figure 1.** Location of Lake Mackenzie overlaid with Open Street Map (OSM) standard map. A regional map highlighting the location of Lake Mackenzie is displayed (**left**), together with the fire boundary and surrounds (**right**).

By combining vegetation and wildfire records from the TasVeg 3.0 provided by ListMap [22], the distribution of the flammability of vegetation can be displayed, as well as potential ecological damage, identified in this work as "sensitivity". Flammability is defined as the fire probability, depending on weather conditions, such as precipitation and wind [23]. In this prototype, flammability is taken into account as a configuration option for prediction of fire spread. On the other hand, sensitivity is a measurement of ecological impact [23]. Flammability in most (63.52%, 204.21 km$^2$) of the burnt area of this wildfire, was classified as Moderate (M) with 20.23% (65.02 km$^2$) classified as High (H). Note that the lost vegetation area is taken as the sum of areas fully or partially overlapping

with the actual burnt area (See figure in Appendix: Proportion of Flammability of Vegetation in Lake Mackenzie Road Fire).

The ecological impact of the fire is assessed by considering the sensitivity of the burnt regions. It is clear from such an analysis that the ecological loss was quite significant in these areas. In fact, the sensitivity of the majority of the lost vegetation is either High (H) (45.27%, 145.54 km$^2$) or Very High (VH) (23.92%, 76.90 km$^2$) (See figure in Appendix: Proportion of sensitivity of vegetation in Lake Mackenzie Road Fire).

## 2.2. System Specifications

The specifications of the prototype system are presented in Table 1. Ubuntu 16.04 is a Linux based operating system [24]. The programming language used is python, running on the python framework, GeoDjango, the spatial web application framework powered by Django [12]. PostGIS is an extension of the relational database management system (RDMS) known as PostreSQL, which is used to predict of fire propagation [25,26]. All data addressed in this section are stored in PostGIS, accessed through GeoDjango. WindNinja is used to produce wind grids consistent with topography. Lastly, Quantum GIS (QGIS) is an open source application for GIS and the versions used in this prototype are 2.18 and 3.2.2 [27]. Note that the former version of QGIS was the latest stable at implementation and the latter was released later and used for better representation.

**Table 1.** System information of the prototype.

| Type | Software/System | Version |
|---|---|---|
| Operating System | Ubuntu | 16.04 LTS |
| Programming Language | Python | 3.5 |
| Python framework | GeoDjango | 2.0 |
| Database Management System | PostgreSQL | 10.0 |
| Spatial Database Extension | PostGIS | 2.4.2 |
| Software | WindNinja | 3.3.0 |
| Software | QGIS | 2.18/3.2.2 |

## 2.3. System Architecture

The system architecture is shown in Figure 2. Firstly, the parameter data for fire models, such as FDIs, are collected from data providers, such as the Australian Bureau of Meteorology (BoM) and Department of Primary Industries, Parks, Water and Environment (DPIPWE). Then, these data are converted and imported into a geo database, such as PostGIS. These data in the database are ingested by fire models running on GeoDjango to predict the fire propagation. Lastly, the prediction is displayed in QGIS in this prototype, and on web browsers in future prototypes.

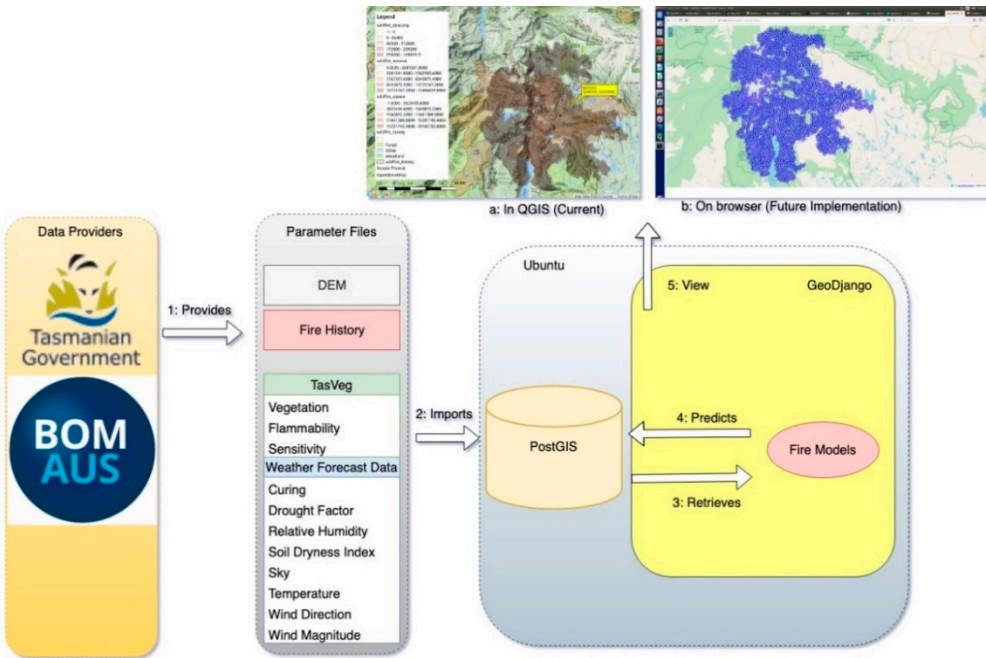

**Figure 2.** System Architecture.

## 3. Methodology

### 3.1. Fire Danger Indices (FDIs)

The FDI or fire danger index has been used in fire weather forecasting across Australia for several decades. For instance, the Australian Bureau of Meteorology (BoM) employs FDI for the operational prediction of fire danger [28]. Three types of FDI are currently calculated operationally: McArthur Grassland Fire Danger Index (GFDI), Forest Fire Danger Index (FFDI), and Button Grass Moorland Index (BGML) [29,30]. On the one hand, FFDI and GFDI are likely to be replaced with more contemporary alternatives in the future, such as Cheney's Dry Eucalypt Forest Fire Model and CSIRO Grassland Fire Spread Model for fire prediction [30]. On the other hand, both GFDI and FFDI are used in this prototype because these are still widely used operationally, while the above alternative models do not explicitly ingest fuel loads [30,31]. *Gymnoschoenus sphaerocephalus*, commonly known as Buttongrass, is common in Tasmania and the associated rate of fire spread is moderate or high [32]. BGML is a model designed specifically for buttongrass, and accounts for fuel load through the age of the buttongrass, and for fuel availability through the parametrization of antecedent rainfall [30]. These indices can be used for the prediction of rate of spread of fire and difficulty of suppression of any fires that are ignited [29]. In this implementation, the FDI class is created as an abstract class, following the concepts of Object-Oriented Programming (OOP). Sub-classes, namely, FFDI, GFDI and BGML, inherit all attributes and methods from FDI. FDI attributes and methods can be overridden by those of the sub-classes (see figure in Appendix: Class diagram: types of FDI). For instance, both FFDI and GFDI have an "index" attribute because the behavior of the "index" attribute varies between sub-classes. The details of each FDI are described in Appendix section: Fire Danger Indices.

#### 3.1.1. Identification of FDI among Vegetation Groups

In order to choose which FDI is used, each FDI is mapped in the vegetation community table in the configuration file. Table 2 shows the default setting of FDIs against vegetation communities.

**Table 2.** Default FDI in Vegetation Community. GFDI, Grassland Fire Danger Index; Forest Fire Danger Index; Button Grass Moorland Fire Index.

| Vegetation Community | Default FDI |
|---|---|
| Saltmarsh and wetland | GFDI |
| Scrub, heathland and coastal complexes | GFDI |
| Highland treeless vegetation | GFDI |
| Rainforest and related scrub | FFDI |
| Dry eucalypt forest and woodland | FFDI |
| Wet eucalypt forest and woodland | FFDI |
| Non-eucalypt forest and woodland | FFDI |
| Moorland, sedgeland, rushland and peatland | BGML |
| Agricultural, urban and exotic vegetation | GFDI |
| Native grassland | GFDI |
| Other natural environments | GFDI |

### 3.1.2. Fuel Load

The default value of a forest fuel load is 12.5 tons per hectare in the operational calculation of FFDI by Bureau of Meteorology (BoM) [33]. In reality, however, fuel load varies. Therefore, fuel load can be configured, and default values are as follows, by assuming that fuel load in eucalypt forests is higher than the average ones provided by BoM and the fuel load in others is lower than BoM average (Table 3).

**Table 3.** Default fuel load in vegetation community.

| Vegetation Community | Default Fuel Load |
|---|---|
| Saltmarsh and wetland | 1.5 |
| Scrub, heathland and coastal complexes | 1.5 |
| Highland treeless vegetation | 1.5 |
| Rainforest and related scrub | 7.0 |
| Dry eucalypt forest and woodland | 14.0 |
| Wet eucalypt forest and woodland | 14.0 |
| Non-eucalypt forest and woodland | 14.0 |
| Moorland, sedgeland, rushland and peatland | 3.0 |
| Agricultural, urban and exotic vegetation | 1.5 |
| Native grassland | 1.5 |
| Other natural environments | 0.0 |

In order to reflect the dependence of flammability on fuel load, the flammability in the vegetation can be multiplied by the above figures as an optional configuration. Flammability is defined as frequency of fire, as mentioned above, and is specified for each vegetation community (See Appendix section: Data Structure) [23]. Therefore, the flammability can vary between vegetation communities to aid the calculation of fuel load more intuitively than by simply using the fuel load of the communities. The coefficients for each flammability type are configurable and the default values are as follows (Table 4):

**Table 4.** Default flammability weight.

| Flammability | Default Weight |
|---|---|
| VH | 2.0 |
| H | 1.5 |
| M | 1.0 |
| L | 0.5 |
| N | 0 |

For example, if the prediction cursor is located in the vegetation community "Native grassland" and flammability is "L", the final fuel load is Fuel Load × Flammability. That is, $1.5 \times 0.5 = 0.75$.

### 3.1.3. Other FDI Configurations

There are some configurations (See Appendix Table: FDI Configuration) other than vegetation community for FDI (Table 2). In particular, curing and drought factors can cause mathematical errors if their values are −1, indicating a missing value. Therefore, the FDI function replaces these values with "WILDFIRE_MISSING_CURING" and "WILDFIRE_MISSING_DF", respectively, in the configuration table with an anomaly including −1.

### *3.2. Prediction of Fire Propagation*

There are two types of grids for predicting fire propagation in this implementation. One is a regular grid, and the other an irregular grid. The general example of the regular grid is a raster described as a tessellation, and the representation of the grid cell is coordinated with pixel size and the scale or resolution [34,35]. However, a regular vector grid instead of a raster grid is created in this prototype so as to contain various types of information for recording fire spread. Another type of grid is an irregular grid, such as Delaunay Triangulation and Voronoi tessellations. A Voronoi diagram is generated with a set of points, such as $p_0$, $p_1$, $p_2$, $\cdots$ $p_{n-1}$ and has more than one nearest neighbor. On the other hand, Delaunay Triangulation is created with three vertices of set P without crossing over each side of the triangulation [36]. These irregular grids can have various orientations, and, therefore, the sizes are also varied. For comparison purposes, all types of grids for prediction inherit the Prediction Vector class and each grid has another vector class, "CentroidPoint", which is a Point Vector and indicates the centroid of each prediction grid. The centroid is used to compute geometry, such as the distance and angle, between grids (See figure in Appendix: Class diagram: prediction grids).

### 3.2.1. Status of Fire

In the prediction, two attributes indicate the fire status (See figure in Appendix: Class diagram: prediction grid). One is "elapse" and the other is "assess". The attribute "elapse" is designed to contain how many seconds the fire takes to reach the grid following ignition. For example, if the fire takes place in the flammable grid, then the elapse is 0 in this grid. If it takes 60 seconds to reach the flammable grid after ignition, the value of the elapse is 60. If the grid is neither flammable nor in a place that the fire can reach, the value is calculated as −1. If the grid has not been estimated, the value is initialized as "None" (Table 5). With regard to "assess", it is "Not Yet" (NT) if either the grid or its adjacent grids have not been estimated. If the elapse is calculated in the grid, then the assessed status is "work in progress" (WIP). If the grid and all its adjacent grids have been estimated, the status is "done" (DN) (Table 6).

**Table 5.** Elapse.

| Value | Description |
|---|---|
| None | The grid has not been estimated yet. |
| −1 | The grid has already been estimated as a non-flammable area. |
| ≥0 | The grid has been estimated as flammable and the value indicates how many seconds the fire is estimated to take in order to reach to this grid from the ignition point. |

Elapse indicates how many seconds the fire will take to reach current grid from the ignited grid.

**Table 6.** Assess status.

| Value | Description |
|---|---|
| NY | Not yet. The grid has not been estimated yet. |
| WIP | Work in progress. The grid is tentatively being estimated as the fire progresses. However not all of its neighbors have been estimated yet, therefore the elapse of this grid can be replaced with a smaller value derived from its neighbors. |
| DN | Done. Both grid and adjacent grids have been estimated. |

Assess indicates a progress status in each grid.

Each prediction grid is stored in the database as a table, such as wildfire_delaunay, wildfire_square and wildfire_voronoi, and accessed via a prediction class, such as Delaunay, Square and Voronoi. In terms of the ongoing fires, grids that have already been estimated themselves—but not all adjacent grids have been estimated yet—can be retrieved using "assess", allowing resumption of prediction even if the prediction process has ceased (See figure in Appendix: Class diagram: prediction grids). Given a long time to execute the prediction of fire, the combination of assess and elapse is useful to continue from where execution previously halted (Section 3.2.3). There are some differences from the simulation by Johnston et al. [15]. This predecessor has two statuses for fire: ignited and unburnt. There are, on the other hand, three statuses, such as NY, WIP and DN in this prototype, as mentioned above. Because of WIP, there are more opportunities to optimize the "elapse" which is another difference and a new additional attribute in this prototype.

### 3.2.2. Data Table for Prediction

In relation to the prediction data table, all polygon shapes, such as Delaunay, Square and Voronoi, have the same extent (See figure in Appendix: Extent of prediction). The range must be able to contain that of actual fire data about the Lake Mackenzie Road Fire. At the same time, the extent of prediction should be a subset of that for the FDI parameters, such as climate, vegetation and DEM, so that the fire travel time can be calculated. The grid resolution is categorized into fine, medium and coarse in the same extension (Table 7). A fine level grid is generated based on 300,000 random points, a medium level grid is based on 75,000 and coarse level grid is based on 12,000.

**Table 7.** Three resolutions in three grid shapes.

| Resolution | Grid Type | Description | Total Number of Grids |
|---|---|---|---|
| fine | Regular | Area size is 60 m². | Square: 300,000 |
| | Irregular | The number of random points is 300,000 in which minimum distance is 45 m | Delaunay: 600,000 Voronoi: 300,000 |
| medium | Regular | Area size is 120 m² | Square: 75,000 |
| | Irregular | The number of random points is 75,000 in which minimum distance is 90 m | Delaunay: 149,971 Voronoi: 75,000 |
| coarse | Regular | Area size is 300 m² | Sare: 12,000 |
| | Irregular | The number of random points is 12,000 in which minimum distance is 225 m | Delaunay: 23,973 Voronoi: 12,000 |

There are several steps to create prediction tables (See figure in Appendix: Overall prediction flow).

First of all, the extent is determined as described above. Then, the grid resolution is determined. For instance, fine grid produces squares with resolution 60 m × 60 m. Then the number of square grid cells is calculated as:

$$x = \frac{456,000 - 420,000}{60} = \frac{36,000}{60} = 600 \tag{1}$$

$$y = \frac{5,400,000 - 5,370,000}{60} = \frac{30,000}{60} = 500 \tag{2}$$

where x indicates the number of grids along the horizontal axis and y represents those along the vertical axis. Then 300,000 records for square polygon are created by QGIS as a wildfire_square. After that, the centroids of square are computed as wildfire_centresquare. The square polygons and their centroids are displayed in Appendix figure (Square and centroid around Lake Mackenzie—fine).

Using the same number of cells as the square polygons, the random point records are created in the same extent as those for square using QGIS (See figure in Appendix: Random point around Lake Mackenzie—fine). The minimum distance between points is specified as 45 m.

In the Delaunay case, the polygons are created as wildfire_delaunay based on the randomized points. After that, the centroid is created as wildfire_centredelaunay (See figure in Appendix: Delaunay and Centroid around Lake Mackenzie—fine).

In the same manner, the wildfire_voronoi is created based on the random points. Then, the centroids for the Voronoi polygons are created (See figure in Appendix: Voronoi and Centroid around Lake Mackenzie—fine).

Medium and coarse grid layers are created following the same procedures in order to analyze the impact of grid size (See figures in Appendix section, Prediction Grid).

### 3.2.3. Prediction of Fire

The figure in Appendix: Pseudo code shows the simplified process of the prediction of fire propagation with the pseudo-code. The prediction starts from the ignition point in the configuration (Table 8). Because the ignition point is the start of the fire, the elapsed time is zero and the assess status is "WIP". Even though the prediction is ceased, the system allows the prediction to take up where it left off last time by retrieving the youngest elapse with the assess status, "WIP" (See figure in Appendix: Cursor movement on prediction grid). After prediction starting or restarting, the immediate neighbor grids are retrieved. The time taken since ignition to reach each neighbor grid is then estimated. If the time taken has already been estimated and the new estimation is smaller, meaning a faster fire progress, then the new estimation supersedes the old one (See figure in Appendix: Estimation of immediate neighbors). Once all immediate neighbors have been estimated, the assess status is updated as "DN". Then, the next youngest estimated polygon, for which the assess is WIP, is picked up from the database (See figure in Appendix: Cursor movement on prediction grid). Note that this prototype is capable of prediction with a single fire ignition only.

**Table 8.** Configuration for fire prediction.

| Configuration Key | Description | Default Value | Note |
|---|---|---|---|
| ignition | Starting place and time to predict | 'x': 439,700, 'y': 5,387,000, 't': "2016-01-19 06:00:00.000000+1100" | |
| maxSeconds | How many seconds to execute prediction | $(60 \times 60 \times 24)$ | seconds |
| maxAreaRatio | How much ratio of area to execute prediction | (1.0) | 0.0 to 0.1 |
| WILDFIRE_ESTIMATE_CONCURRENT | Concurrent process for neighbor estimation | True | |
| WILDFIRE_ESTIMATE_DIRECTDB | Stored procedure can be used to retrieve raster data | True | |

It is possible to configure the ignition place and time into the configuration file. The prediction is automatically stopped when all grids are made "DN" or the predicted area size becomes equal to or greater than the actual burnt area. In addition, the prediction can stop when the elapse on the current prediction exceeds the maximum number of seconds, "maxSeconds", in the configuration file (Table 8; figures in Appendix: Flowchart—prediction and Flowchart—estimate time ingesting

parameters and selecting FDI). This is useful to enable the scheduling of the execution of prediction. For instance, the user can stop the prediction after running 12 h for maintenance purposes when configuring maxSeconds as $(60 \times 60 \times 12)$. Afterwards, the program can resume where it left last time by increasing maxSeconds as, for example, $(60 \times 60 \times 24)$.

In order to improve the process speed, some contrivances are necessary. Although the process of prediction is carried out sequentially, the estimation for adjacent grids from certain grids can be concurrently performed because the elapse on each adjacent cell from the current grid is independent of other adjacent grids (See figure in Appendix: Flowchart—prediction). For instance, a square grid usually has eight adjacent grids and none of them influence other neighbors until the current, i.e., the central grid, finishes calculation and is marked "DN" (See figure in Appendix: Adjacent grids). In other words, adjacent grids are computed from only the current grid at this stage. Therefore, this prototype has configuration, "WILDFIRE_ESTIMATE_CONCURRENT", to switch to a concurrent thread calculation for the immediate neighbors (Table 8). Note that the neighbors' elapses are tentative estimations at this stage, and, therefore, their assess statuses are "WIP", as mentioned above (Table 6). Another efficiency can be made in retrieving DEM and climate data from the database. Although this prototype employs python with GeoDjango, the speed would be compromised if these values were retrieved sequentially. Therefore, values can be retrieved simultaneously from the database when the configuration, "WILDFIRE_ESTIMATE_DIRECTDB" is True (Table 8).

### 3.3. Verification, Validation and Acceptability of Model

In general, there are a few terms employed to measure the quality of models, such as verification, validation and acceptability (Table 9). In verification, the expected results need to be known beforehand. Verification is conducted by comparison of expected with actual results. Validation is employed to evaluate the degree to which the model represents phenomena in the real-world. Acceptance is part of the decision-making process to assess the quality of model by verification, validation and user bias [37].

**Table 9.** Verification, validation and acceptability.

| Term | Description |
|---|---|
| Verification | Evaluation of the discrepancy between the expectation and actual result |
| Validation | Evaluation of the gap between actual result and real-world |
| Acceptability | Decision-making of acceptability of verification and validation |

In this project, only verification is employed, due to it being academic research; output is compared with the fire history provided, assuming the latter as the actual result. However, the discrepancy between this wildfire history and the real-world is not validated in this prototype. To perform verification, a confusion matrix is used to compare the actual result with the prediction.

### 3.3.1. Confusion Matrix

The concept of the confusion matrix originated from machine learning and is designed to classify the frequency of various statuses against certain behaviors and summarize accuracy and precision by comparison of predictions with observed results [38–41]. Details of the confusion matrix are described in the Appendix section: Confusion Matrix.

### 3.3.2. Other Common Criteria for Data Quality

In GIS, several common criteria are used to assess quality of source data, such as completeness, currency and applicability [35,42].

First of all, the indicator of both temporal and spatial coverage of the study area is completeness. The gridded forecast weather data, such as SDI and temperature, provided by BoM, covers from 18th

January until 3rd February 2016 (section in Appendix: Data Structure). The duration of topographically dynamic wind data is the same as that for forecast grid data above. In respect to the spatial coverage, the extent of the prediction area encloses the fire history so that the additional area allows the measurement of the prediction area even if the prediction shows a different track from the observed fire area. In addition, almost all the prediction areas are contained by the parameter data, such as DEM, TasVeg and the weather forecast grid (Section 3.2). Secondly, compatibility states how well the data can be switched between different scales. Thirdly, consistency indicates how close the conditions are between separated crude data. For instance, if there are two DEM files in which resolutions and creators are different, then the consistency is compromised. Fourthly, currency is the indicator of whether or not the data is updated. Lastly, applicability indicates how suitable the produced results are for the model. In this prototype, applicability shows which combination of shape and grain size is the most appropriate for the fire prediction.

## 4. Results

This section addresses both the expected and actual results of the execution of the fire prediction. The prediction was executed on each polygon type and on each grain size layer until the total predicted area became equal to or larger than the real burnt area. In the prototype, the fire was assumed to ignite at 6 am on 19th January 2016 in local Tasmanian time, the coordinate is (x = 439,700, y = 5,387,000) in GDA94 MGA zone 55 mentioned in the configuration (Table 8; section in Appendix: Results).

### 4.1. Expected Result

The Delaunay polygon in the fine grid layer was expected to represent the shortest path, that is, to have the least elapse when the total area reached the same or greater than the real fire. Thus, the finer the grid is, the better it reflects the fire spread and the shorter the paths it displays. This would hold true for each of the three types of polygons, Delaunay, Square and Voronoi. That is, Delaunay is the smallest size among the three types at each granularity (See figure in Appendix: Comparison of average area size). Another reason for this anticipated result follows from the number of adjacent grids. Because Delaunay has the greatest number of adjacent grids, averaging approximately 12.37 compared to Square (7.94) and Voronoi (5.96), each neighbor has approximately 12 chances to recalculate the tentative elapse at maximum. That is, Delaunay has more opportunities to retrieve the least time than the others do (See figure in Appendix: Comparison of number of immediate neighbors).

### 4.2. Actual Result

The actual results partially appear as expected. However, some aspects of the results did not meet expectations (See section in Appendix: Results). With regard to the execution of prediction, there are nine predictions and each prediction was conducted until the total size of the burnable area, where the grid is assessed as "DN" and elapse > 0, is equal to or greater than the Lake Mackenzie Road Fire record, which is approximately 247.11 km$^2$. In this section, the results of execution and their tendencies are addressed at first. Then the result is analyzed using the confusion matrix. Lastly, the results are summarized against the five criteria for data quality, that is, applicability, compatibility, completeness, consistency and currency.

#### 4.2.1. General Tendency

In terms of the elapsed time, which indicates how long it takes the modelled fire to propagate from the ignition point to the current grid, Delaunay is the quickest among three polygon types, followed by Square and Voronoi (See figure in Appendix: Elapse—actual). On the other hand, elapsed time does not appear to change in proportion to granularity. Despite the expectation that the medium size would have the second shortest elapsed time among the grain sizes, it actually shows the shortest time (See figure in Appendix: Elapse—actual).

With regard to matching the size of the actual fire and predicted burnt area, there is no significant relationship between the types of polygon or size of grids. For instance, the coarse Delaunay grid has the largest shared area size (52.67%) among the nine results while the coarse grids in Square (47.35%) and Voronoi (46.63%) follow the medium grid in each polygon, that is, 48.17% and 48.35% respectively. Interestingly, the fine grids in Square (46.59%) and Voronoi (43.05%) demonstrate the least matching in each shape (Table 10).

**Table 10.** Overlapped area (percentage).

| Polygon | Grain Size | Shared Area with Actual (%) | Shared Area with Actual (km$^2$) |
|---|---|---|---|
| Delaunay | fine | 46.30 | 114.40 |
| | medium | 46.16 | 114.06 |
| | coarse | 52.67 | 130.16 |
| Square | fine | 46.59 | 115.13 |
| | medium | 48.17 | 119.03 |
| | coarse | 47.35 | 117.00 |
| Voronoi | fine | 43.05 | 106.37 |
| | medium | 48.35 | 119.47 |
| | coarse | 46.63 | 115.23 |

### 4.2.2. Result with Confusion Matrix

The verification from the confusion matrix is presented in Table 11.

**Table 11.** Verification from confusion matrix by sizes, fine, medium, coarse, as well as shapes, such as Delaunay (D), Square (S) and Voronoi (V).

| | Fine | | | Medium | | | Coarse | | |
|---|---|---|---|---|---|---|---|---|---|
| | **D** | **S** | **V** | **D** | **S** | **V** | **D** | **S** | **V** |
| **True Negative** | 201,055 | 101,446 | 101,610 | 50,591 | 25,152 | 25,455 | 8383 | 4013 | 4204 |
| **False Positive** | 8856 | 3034 | 3386 | 1983 | 968 | 841 | 44 | 187 | 24 |
| **False Negative** | 261,617 | 129,911 | 129,760 | 65,110 | 32,687 | 32,406 | 10,139 | 5241 | 5069 |
| **True Positive** | 128,432 | 65,609 | 65,244 | 32,287 | 16,193 | 16,298 | 5407 | 2559 | 2703 |
| **Total** | 599,960 | 300,000 | 300,000 | 149,971 | 75,000 | 75,000 | 23,973 | 12,000 | 12,000 |
| **Accuracy (%)** | 54.92 | 55.69 | 55.62 | 55.26 | 55.13 | 55.67 | 57.52 | 54.77 | 57.56 |
| **Misclassification Rate (%)** | 45.08 | 44.32 | 44.38 | 44.74 | 44.87 | 44.33 | 42.48 | 45.23 | 42.44 |
| **Precision (%)** | 93.55 | 95.58 | 95.07 | 94.21 | 94.36 | 95.09 | 99.19 | 93.19 | 99.12 |
| **Specificity (%)** | 95.78 | 97.10 | 96.78 | 96.23 | 96.29 | 96.80 | 99.48 | 95.55 | 99.43 |
| **Prevalence (%)** | 65.01 | 65.17 | 65.00 | 64.94 | 65.17 | 64.94 | 64.85 | 65.00 | 64.77 |
| **True Positive Rate (%)** | 32.93 | 33.56 | 33.46 | 33.15 | 33.13 | 33.46 | 34.78 | 32.81 | 34.78 |
| **False Positive Rate (%)** | 4.22 | 2.90 | 3.22 | 3.77 | 3.71 | 3.20 | 0.52 | 4.45 | 0.57 |

Firstly, although there is no significant trend in total, the fluctuation in coarse polygons is worth addressing. Among the coarse polygons, Voronoi indicates the highest accuracy (57.56%) and therefore its inverse rate, that is, low misclassification rate becomes the lowest (42.44%). On the other hand, the Square in coarse shows the opposite trend, with the lowest accuracy (54.77%) and

highest misclassification rate (45.23%). This trend holds true for other indicators, such as specificity, true positive rate and false positive rate. On the other hand, other granularities do not show significant differences across the indicators.

Secondly, the precision and specificity level off around 95% (Table 11). These levels occur because of the small FP, representing the ratio of the actual result being false, while the prediction is true. For instance, the equation of precision is represented as TP/(TP+FP) and specificity is TN/(TN+FP). Then, the result of calculation becomes close to 100% because of the small ratio of FP, which means that it is rare that the actual result shows false while prediction is true. In the same manner, FP causes a low false positive rate, the equation for which is (1-specificity). Again, the precision shows how close the values of the estimated data are to each other, while the accuracy indicates how close the observed and predicted values are. The high precision throughout the shapes and granularities of prediction arose under the same circumstances. That is, these employed the same FDI and digested FDI's parameters, such as classification of vegetation, curing, temperature, wind direction and magnitude [42].

Lastly, the prevalence, the equation for which is (FN+TP)/(TP+TN+FP+FN), levels off around 65%. Here FN indicates the case where false is predicted while true is observed and TP represents the ratio where true is predicted and observed. This appears sensible because the prediction stops when the area size is equal to or larger than observed fire area. Therefore, the ratio of the evaluated area becomes similar to that which is not evaluated, regardless of the content of the result. Consequently, Delaunay is the most appropriate for the estimation of elapse among three types of polygon and the medium granularity is recommended by considering the expense and quality of matching area (Section 5).

### 4.2.3. Verification with Other Common Criteria

There are several indicators for data quality, as mentioned above (Section 3.3.2). Firstly, both spatial and temporal data coverages are addressed as completeness. In terms of spatial coverage, the prediction has completed within the extent of each prediction polygon (Table 9). On the other hand, there is a shortage of days in the parameters, which range from 18th January until 3rd February 2016. For instance, several predictions, such as fine Square, coarse Square and all Voronoi, overran this temporal coverage (Section 5.3). The issue of the temporal coverage is, however, limited, as mentioned later on (Section 6). Secondly, all crude data are consistent throughout all three sizes of prediction because all grain sizes and polygons ingest the same source data (See Appendix section: Data Structure). In addition, the same configuration file, such as the selection of FDI and fuel load, is also referred to by each grain size and polygon of prediction (Section 3). Thirdly, there is a lack of consistency because the resolution varies and is addressed as a prototype limitation (Section 6). Fourthly, the created or modified date of each source datum in Table 12 do not meet this criterion well for temporal consistency [22,43–45]. For instance, TasVeg 3.0 was created in November 2013. Therefore, the vegetation data was not entirely current in January 2016. This issue is also addressed as a limitation. Lastly, the applicability is discussed in Section 5.

**Table 12.** Data source and date.

| Dataset | Last Modified |
| --- | --- |
| History | 07-09-2017 |
| TasVeg 3.0 | 11-11-2013 |
| DEM | 17-11-2017 |
| Forecast weather | |
| CuringRF | 26-01-2016 |

## 5. Discussion

Here we discuss the tendencies observed in the results of this prototype between polygon shapes and sizes. In addition, we analyze the impact of each parameter on FDIs and then make some recommendations.

### 5.1. Geometric Data and Their Impact on the Prototype

#### 5.1.1. Polygon and Elapse

Firstly, the more adjacent polygons the grid has, the lower the elapsed time from ignition to completion of the predicted burn. For instance, Delaunay has the greatest number of immediate neighbors, approximately 12, followed by Square and then Voronoi, with approximately 8 and 6, respectively. The elapsed time of Delaunay is the lowest, followed by Square and Voronoi. On the other hand, the granularity is not proportional to the increase of elapsed time. In fact, the medium-sized polygon shows the least elapse across the polygon types (Section 4.2). The attribute elapsed time is not contained in each cell in the predecessor's simulation by Johnston et al., but implemented in this prototype [15]. Consequently, Delaunay is found to be the most accurate because of the most evaluated chances regardless of size.

#### 5.1.2. Polygon Size, Accuracy and Precision

The second trend is that the medium size of polygon shows the most stable and precise of all polygons in the confusion matrix (Section 4.2). That is, indicators such as accuracy, precision, specificity and prevalence are the closest to each other in the medium granularity among the three resolutions. Fine and coarse granularities are less close. Figure 3 shows the range between the minimum and maximum indicator values in polygon types.

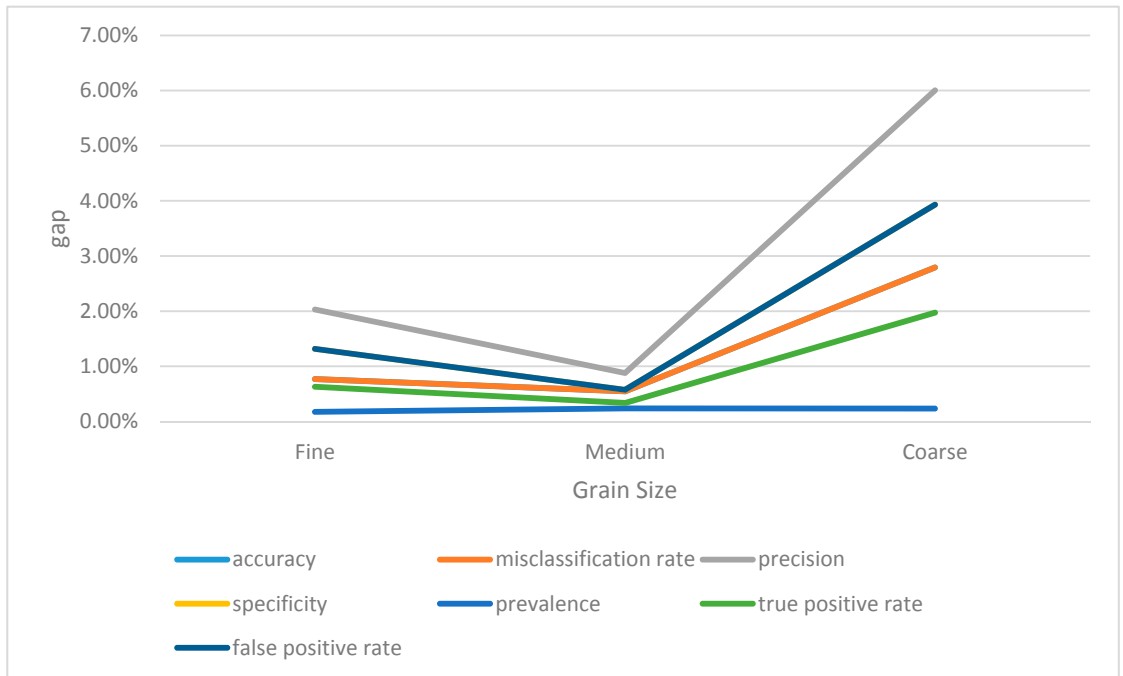

Note: Gap between maximum and minimum values in specificity tracks the same line as false positive rate because the equation of specificity is (1—false positive rate). Hence the line for specificity is hidden by false positive rate in the graph.

**Figure 3.** Confusion matrix indicator gap.

The question arises as to why the middle grain size shows the greatest precision, as mentioned above. The potential reason is because of the representativeness against the real world. The map generalizes the object in the real world [42] and entities should be represented in several grids [34]. With regard to the coarse grid, the area sizes of both Square and Voronoi are close to the resolution of wind direction and magnitude output from WindNinja, and entities on the borders of the grids might be distorted. Therefore, the coarse grid size is not ideal to represent fire propagation. On the other hand, the medium polygons are well contained in the grids of the main parameters of prediction, such as CuringRF, WindMagWN and WindDirWN (See Table 7; table in Resolution by raster type in Appendix). For example, the WindMagWN and WindDirWN are about six times as large as the middle grid of Square and Voronoi grains.

$$w \div mS = 308.68^2 \div 120^2 \fallingdotseq 6.62 \tag{3}$$

where w is either WindMagWN and WindDirWN, and mS indicates the middle Square.

The fine level, however, is unnecessarily small in comparison to these wind parameters (Figure 4).

$$w \div sS = 308.68^2 \div 60^2 \fallingdotseq 26.47 \tag{4}$$

where sS denotes a small Square. Note that the DEM grain size is substantially smaller than the size of any polygon grain, i.e., $25.00^2 = 625$ (See Figure 4 and table in Appendix: Resolution by raster type). As such, any impact is marginal, unless the topography drastically fluctuates, because the slope calculated from one (not its shape but its) centroid to another in DEM is averaged when calculated. Thus, the location of the centroid has an impact on the calculation. For example, inclination is calculated as the difference in level between two centroids, and the shape of polygons is irrelevant. If there are two centroids in the same DEM grid cell, then the inclination is 0°, i.e., flat. On the other hand, the inclination can vary if one centroid is located in a different grid cell of the DEM.

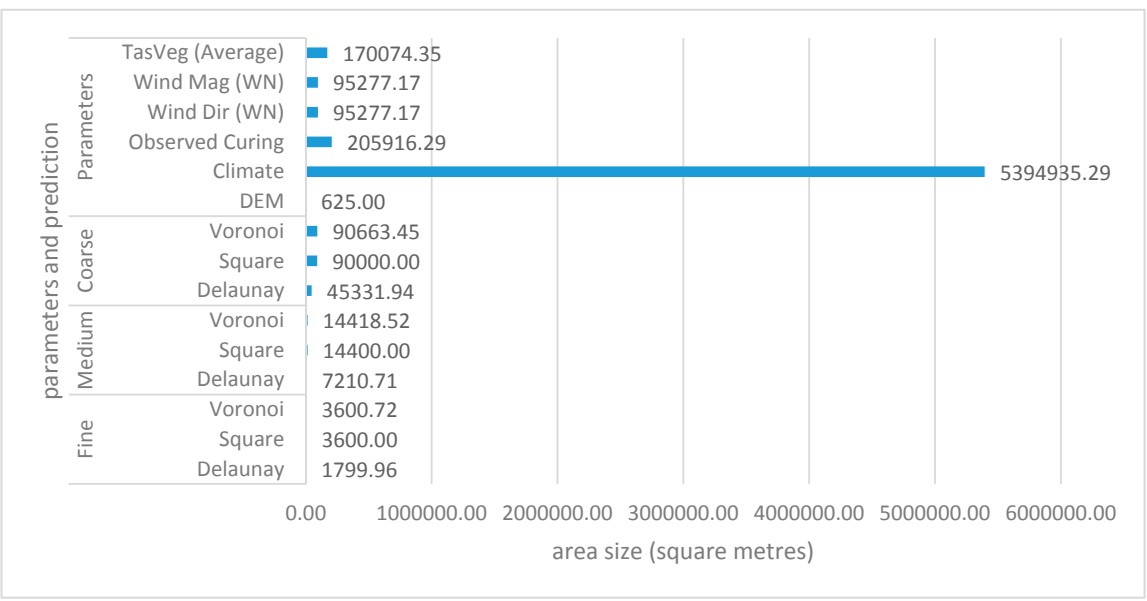

**Figure 4.** Area sizes of prediction and parameters.

The impact of the difference between regular and irregular shapes is limited. Although the process of creating grids is different between regular, that is, square, and irregular grids, such as Delaunay and Voronoi, the grids are well-distributed by the minimum distance between the randomized points. For example, the centroid of each irregular polygon is generated based on random points; however, these random points are constrained by the minimum distance between centroids, i.e., 45 m in the

fine grain, 90 m in the medium size and 225 m in the coarse grain (Section 3.2.2). This phenomenon occurs because the centroids were created based on a constrained pseudo-randomness. For instance, each randomized point is ensured to be at least 45 m away from the closest point in the fine grain size. Hence, the impact of this uncertainty is also limited. In addition, the fire distortion in cell-based method [15] can be successfully suppressed in this prototype due to the centroid based calculation.

## 5.2. Spatial Data and Their Impact on the Prototype

Topographical data, i.e., the DEM, has a significant impact on the fire behavior. Although both slope and wind have been considered as major factors in the progression of a fire, the wind is affected by the terrain [18,46]. In particular, wind direction and magnitude are re-calculated by WindNinja. On one hand, weather data have high temporal variability and change more rapidly than other data types, and it is, therefore, more difficult to manage the quality of wind data. On the other hand, terrain data are static and vegetation is low temporal variability data, which suggests that the management of vegetation through, for example, prescribed burning [47], in particular, might have a potential to mitigate fire spread as an environmental management approach.

## 5.3. Recommendation

From a cost aspect, the medium grain is most appropriate to predict fire propagation. Table 13 shows the cost-effectiveness where cost effectiveness is defined as:

$$cost-effectiveness \; = \; \frac{execution time}{elapse} \tag{5}$$

**Table 13.** Cost effectiveness.

| Grain Size | Polygon | Execution Seconds | Execution Time (hh:mm:ss) | Elapse Seconds | Elapse (dd, hh: mm:ss) | Cost-Effectiveness (%) |
|---|---|---|---|---|---|---|
| fine | Delaunay | 99,176 | 27:32:56 | 958,402 | 11, 2:13:22 | 10.35% |
| | Square | 126,368 | 35:06:08 | 1,394,746 | 16, 3:25:46 | 9.06% |
| | Voronoi | 383,774 | 106:36:14 | 1,682,749 | 19, 11:25:49 | 22.81% |
| medium | Delaunay | 23,332 | 6:28:52 | 889,929 | 10, 7:12:09 | 2.62% |
| | Square | 31,848 | 8:50:48 | 1,155,314 | 13, 8:55:14 | 2.76% |
| | Voronoi | 92,073 | 25:34:33 | 1,579,680 | 18, 6:48:00 | 5.83% |
| coarse | Delaunay | 14,195 | 3:56:35 | 1,042,933 | 12, 1:42:13 | 1.36% |
| | Square | 4692 | 1:18:12 | 1,485,585 | 17, 4:39:45 | 0.32% |
| | Voronoi | 3736 | 1:02:16 | 2,324,237 | 26, 21:37:17 | 0.16% |

This indicator shows how much time the system needs to spend to calculate a prediction after a fire occurrence. Although the coarse grid achieves a fast calculation, such as 1:02:16, 1:18:12, and 3:56:35 for Voronoi, Square and Delaunay respectively, the quality varies between polygon types, as mentioned above (Section 4.2). On the other hand, the fine grid shows a cost-effectiveness of between 9.06% and 22.81%. For emergency management purposes, it is necessary to generate a prediction as soon as possible and the fine grid, therefore, not appropriate because of its slow execution speed. By considering the balance between cost-effectiveness and emergency management requirements, the medium grid is recommended.

Based on the above, a staged forecast approach is suggested. Thus, a coarse prediction can be released as a primary prediction, then a medium prediction follows as a more detailed estimate of fire spread area.

In respect of the selection of FDI and fuel load, vegetation types appear to have a significant impact on the prediction, and it might be useful to create a vegetation base polygon. Because TasVeg 3.0 consists of polygons, it is possible to create a centroid using QGIS (See figure in Appendix: Centroid in red contained in TasVeg). A drawback is the inconsistent size of polygons. For example, the maximum area of the vegetation in the prototype is 272,149,146.93 $m^2$, the minimum is 0.77 $m^2$, and the average is 170,074.35 $m^2$. The solution might be to use "Omitted" as a feature generalisation, in which features smaller than a threshold value are excluded from the map [34].

## 6. Limitations

There are a number of limitations of this prototype. Since limitations of the quality of source data and observed data impinge on the final output, it is essential to address data history [42]. In addition to the data quality, the limitations of algorithms (e.g., FDIs) are also noted.

Firstly, some parameter data are quite coarse. For instance, the resolution of DF, RH and SDI in forecast weather data is approximately 2.32 $km^2$ (See table in Appendix: Resolution by raster type). Even though there might, for example, be a topographic ridge enclosed in a forecast weather grid, the entire area has the same value for each weather parameter, which is not realistic because the duration of sun light will be different between sides of the ridge, affecting a number of weather parameters, such as temperature and curing. In addition, vegetation can also be different over such a large area partly because of topographical variation within the area.

Secondly, the parameters for FDI ended 3rd February 2016. If the prediction had not become as large as the fire history yet, the last available data were reused. Although this may produce a discrepancy from the actual fire history, the remnant days were not during peaks of fire activity, which occurred from 19th until 22nd January (Section 2.1). Therefore, the impact of the lack of parameters is marginal.

Thirdly, TasVeg 3.0 is obsolete because there is more than two years between the modified data, November 2013 and the actual fire, January 2016. However, the vegetation and ecological information were captured before the wildfire and these are still useful as parameters for sensitivity, that is, ecological impact, and flammability estimated from the fire record.

Fourthly, the fire history is not evaluated in this prototype. For example, some water areas are identified as burnt areas in meshed-red while the prediction of fire using fine Delaunay indicated unburnt areas in red (Figure 5). The image (a) shows the Fisher River in the wildfire area and this river was recorded as part of a burnt area in the Lake Mackenzie Road Fire shown in image (b). On the other hand, the prediction in image (c) does not count it as burnt because the water is not burnable in the prototype.

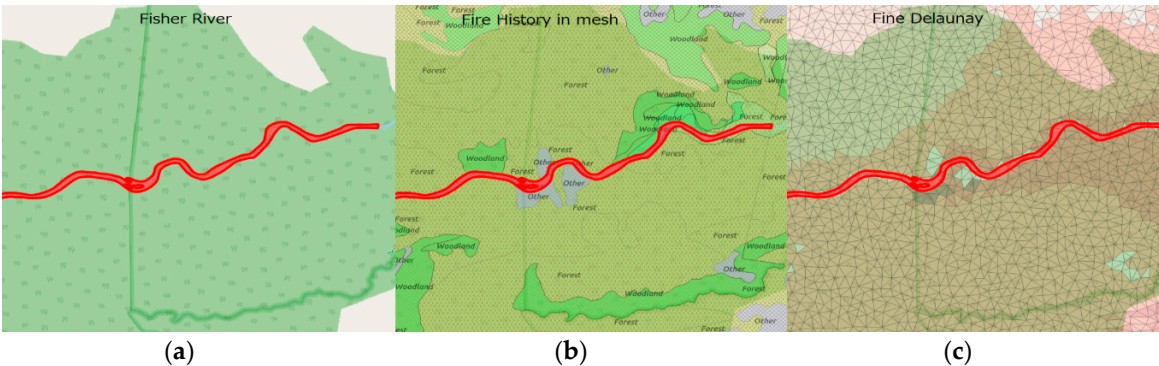

**Figure 5.** Comparison from left to right, (**a**) Fisher River in red, (**b**) Fire History in red mesh, (**c**) Delaunay in red triangle polygons.

Fifth, the temporal accuracy of fire propagation is not evaluated because the ground truth data does not contain sufficient temporal resolution of fire propagation. It is important to evaluate when and where the fire front reaches, and this temporal accuracy needs to be assessed in future implementations.

The Sixth limitation is the inconsistent length of connected sides between grids. This occurs because irregular grids are generated from random points and the side length are naturally varied. Square grids permit easier assessment of differences between neighbor grids than do irregular grids, such as those used in this prototype. However, in a square grid, while perpendicular neighbors have the same length common boundaries, the corner neighbors have no common boundary. This could be solved by use of a hexagonal grid, to be implemented in future work, because all six sides are the same length [48].

Lastly, the fire is assumed to traverse between adjacent grids, that is, immediate neighbors only in this prototype. However, fire is often reported to across an unburnt area, such as road. For example, spotting of up to 33 kilometers was observed in the Black Saturday fires in Victoria, on 7th February 2009 [3].

## 7. Conclusions and Future Work

This study aims to develop a spatially explicit prototype for wildfire prediction. The study has achieved its first objective, identifying the main issue of fire prediction, i.e., distortion of fire direction, when employing a regular grid, such as cellular automata. Then, the study presents a prototype fire prediction with various granularities and polygon shapes where the calculation is based on the centroid of each polygon, to solve the issue of regular grid prediction, which is the second objective. The core predictions of this system are FDIs, such as BGML, GFDI and FFDI. In order to predict a fire using FDIs, a number of parameters are prepared, including Curing, DEM, DF, RH, SDI, Vegetation data, WindDirWN and WindMagWN. It is found that the number of adjacent grids has an impact on the elapsed time from an ignition point to a target grid. A Delaunay grid, for which the average number of adjacent grids is the greatest among the three shapes of polygon investigated, shows the shortest elapse for the spread of fire. With regard to the quality, the medium size of prediction is verified as the best trade-off between cost and time among the three sizes of prediction polygons, while the coarse size shows the best cost-effectiveness. A staged approach, where a coarse size prediction is released at first and a medium sized one follows, can be a practical solution, having the best cost-effectiveness as guidance for the purposes of urgent evacuation. The third objective of the study, development of an efficient evacuation tool, has been accomplished by this staged approach.

For future study and development, fire intensity and the adjacency of grids should be considered, as fire is often reported to cross an unburnt area. Replacement of current, essentially arbitrary, prediction shapes with vegetation polygons should also be considered. Further, the simulation results could be integrated into directional guidance to evacuation trigger points, such as ridgelines, rivers and roads [49]. In addition, the three grain sizes are expected to be dynamically switched depending on the status of the user. For example, from the ignition point to the current area, the coarse grid layer will be used. Then the finer grid will be employed to display the detail of fire propagation around the application user. In relation to accuracy of the predictions, it is necessary to further investigate the best use of cellular automata in simulations, such as those conducted here. Although distortion of the fire shape is recognized as a common issue in other research, it is worth comparing the output with from gridded approaches in modelling large wild fires. In addition, other fire models, such as Dry Eucalypt Forest Fire Model 2012 and CSIRO Grassland Fire Spread [30], also should be selectable as the underlying fire behavior models by configuration for improved accuracy in the output fire predictions. Temporal accuracy of fire propagation should also be. With regard to computational speed, CPU or GPU optimization, as used in other integrated fire simulation systems such as CSIRO Spark [20], should be considered in addition to the core concept of applying a cloud computing approach, from the original research. Other validation methods, such as Kappa coefficients, may be also employed to

examine prediction accuracy [50]. Finally, some parts of this technology may be applied to other types of disaster incidents, such as flooding.

**Supplementary Materials:** The following are available online at http://www.mdpi.com/2220-9964/8/4/194/s1, There are further supporting figures and tables in another file, *Appendix—Dynamic Evacuation Navigation from Wildfire*.

**Author Contributions:** Conceptualization, methodology and analyses, Mitsuhiro Ozaki; writing—original draft preparation, Mitsuhiro Ozaki; writing—review and editing, Mitsuhiro Ozaki, Jagannath Aryal and Paul Fox-Hughes; supervision, Jagannath Aryal and Paul Fox-Hughes.

**Funding:** This research received no external funding.

**Acknowledgments:** I would like to acknowledge the following data and software providers:

- Data

  ○ Department of Primary Industries, Parks, Water and Environment (DPIPWE) through ListMap.

    ■ Digital Elevation Model (DEM)
    ■ TasVeg 3.0

  ○ BoM

    ■ Forecast weather grid, such as Curing, Soil Dryness Index (SDI), Drought Factor (DF), Relative Humidity (RH), Temperature, Wind Direction and Wind Magnitude

- Software

  ○ Canonical Ltd.

    ■ Ubuntu 16.04 LTS

  ○ The PostgreSQL Global Development Group

    ■ PostgreSQL 10.0

  ○ Django Software Foundation

    ■ GeoDjango 2.0

  ○ QGIS Community

    ■ Quantum GIS software (QGIS) 2.18/3.2.2

  ○ U.S. Forest Service

    ■ WindNinja 3.3.0

**Conflicts of Interest:** The authors declare no conflict of interest.

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
