# Peer review of "Dynamic Wildfire Navigation System"

_ijgi, doi:10.3390/ijgi8040194_

Round 1

Reviewer 1 Report

Review # ijgi-481120

General comments

The authors significantly improved their manuscript. However, there are some issues that demand minor revision before publishing to the International Journal of Geo-information, which are related to the language used. I suggest the authors improve the language style before providing a final version of the manuscript.

Specific comments.

INTRODUCTION

1.    Lines 15 – 18. It is not clear what you mean. Please, improve the phrase.

2.    Line 20. I suggest the authors add “is located” before Lake Makenzie.

3.    Lines 46-46. The phrase “are addressed” is appeared twice.

4.    Line 62. I suggest the authors delete “in this prototype”.

5.    Line 72. The fuels are usually characterized as “highly” flammable.

6.    Line 83.  It is not clear what you mean. Please, rephrase.

7.    Line 93. WindNinja reference is missing.

I recommend the authors to check this section carefully and to improve it.

STUDY AREA AND SYSTEM ARCHITECTURE

1.    Figure 1 is not mentioned in the paragraph.  

METHODOLOGY

1.    Line 163. Cruz et al. reference is missing.

2.    Line 166. Please, add the “()…..associated rate of fire spread…()”. 

3.    Line 192. Please add a dot (.) after [23].

3.2.3 Prediction of fire

       Please correct the missing lines (283), (302-303), (318), (325), (369), (406 -407)

 LIMITATIONS

1.    I suggest the authors avoid the Sixth limitation.

Author Response

INTRODUCTION
Lines 15 – 18. It is not clear what you mean. Please, improve the phrase.
Regular and irregular are now differentiated more clearly.

Line 20. I suggest the authors add “is located” before Lake Makenzie.
“The fire study area is located around Lake Mackenzie in the central north of Tasmania …”

Lines 46-46. The phrase “are addressed” is appeared twice.
“At first, the study area and prototype architecture are illustrated, then the methodologies, such as calculation of FDIs, prediction of fire propagation, and verification are addressed.”

Line 62. I suggest the authors delete “in this prototype”.
As advised.

Line 72. The fuels are usually characterized as “highly” flammable.
As advised.

Line 83.  It is not clear what you mean. Please, rephrase.
Now the example is given.

Line 93. WindNinja reference is missing.
The reference is now provided.

STUDY AREA AND SYSTEM ARCHITECTURE
Figure 1 is not mentioned in the paragraph.  
Now the figure 1 is addressed.

METHODOLOGY
1.    Line 163. Cruz et al. reference is missing.
The reference is now provided.

2.    Line 166. Please, add the “()…..associated rate of fire spread…()”.
As advised.

3.    Line 192. Please add a dot (.) after [23].
As advised.

Please correct the missing lines (283), (302-303), (318), (325), (369), (406 -407)
As advised.

LIMITATIONS
1.    I suggest the authors avoid the Sixth limitation.
As advised.

Please see attached manuscript.

Reviewer 2 Report

The authors have made significant changes to this paper; primarily these changes served to reduce its scope and claims, rather than to materially address the concerns of the initial review with additional analyses etc. but this is understandable given the time constraints on this journal's review process. I think the present scope of the analysis, now that the limitations of this as a "prototype" have been emphasized in the text, make this more suitable for publication. I can only hope future work will conduct the additional analyses identified, and obtain the promised processing speed increase over traditional fire behaviour modeling. 

I would request a few minor changes based on this revision; re-reading it, it becomes clear that your methodology is very closely based on that provided by reference 15 (Johnston et. al. 2008) - you do state this, but it seems much of the purpose of your paper is to validate the Johnston methodology,  test alternative geometric configurations etc. so it may be worthwhile enhancing discussion of this paper, eg. in the methods section (how do your methods differ?) and in the discussion section (how does your accuracy compare to other applications of this method in the Johnston paper?).  Your paper makes much more sense in extension and in comparison to this earlier work. 

Also a minor formatting correction - Line 165 - Gymnoschoenus sphaerocephalus, being a botanical species name, should be italicized

Author Response

The authors have made significant changes to this paper; primarily these changes served to reduce its scope and claims, rather than to materially address the concerns of the initial review with additional analyses etc. but this is understandable given the time constraints on this journal's review process. I think the present scope of the analysis, now that the limitations of this as a "prototype" have been emphasized in the text, make this more suitable for publication. I can only hope future work will conduct the additional analyses identified, and obtain the promised processing speed increase over traditional fire behaviour modeling.
If I, Mitsuhiro, proceed to phD course, I might continue developing of this prototype.

I would request a few minor changes based on this revision; re-reading it, it becomes clear that your methodology is very closely based on that provided by reference 15 (Johnston et. al. 2008) - you do state this, but it seems much of the purpose of your paper is to validate the Johnston methodology,  test alternative geometric configurations etc. so it may be worthwhile enhancing discussion of this paper, eg. in the methods section (how do your methods differ?) and in the discussion section (how does your accuracy compare to other applications of this method in the Johnston paper?).  Your paper makes much more sense in extension and in comparison to this earlier work.
The explanations of differences are mentioned in 3.2.1 and 5.1.1.

Also a minor formatting correction - Line 165 - Gymnoschoenus sphaerocephalus, being a botanical species name, should be italicized
As advised.

Please see attached manuscript.

This manuscript is a resubmission of an earlier submission. The following is a list of the peer review reports and author responses from that submission.

Round 1

Reviewer 1 Report

  REVIEW MANUSCRIPT # 454469 - IJGI

General comments

The authors in their article present a prototype system for fire prediction in Australian forest environments. The topic is interesting since it evolves a number of (open utility) sophisticated Software / Systems, along with fire behavior models and dynamic meteorological data as inputs. It is also true that such systems could be valuable during firefighting operations and in-time evacuations. However, there are a number of issues related to the manuscript which require revision in order to increase its scientific merit.  

1.       Development of relevant systems to those described in the article are of great interest for other countries facing also the same problems. Therefore, I recommend the authors to re-write parts of the paper with a view to an international audience. Otherwise, the article in the current form could be more suitable for publication in a regional or national journal rather than an international one, such as the I.J.G.I.  In this context, I propose they introduce a new brief paragraph along with a general flowchart describing the basic parts of the system to show clearly the approach followed for the proposed prototype so as to increase its potential transferability. The strongest part of the paper to my opinion is the system's basic architecture, which should be further highlighted. Therefore, in the new general flowchart, the GIS box could be replaced by QGIS (as recommended), or for example by ArcGIS, MapInfo etc. The inhered fire behavior model (Noble et al. 1980) is connected only to Australia's forest environments, thus restricting its applicability to specific ecosystems. A general FIRE MODEL box will show the option of selecting alternative fire models, as reasonable solutions for other countries too located for example in the Mediterranean Basin or in Canada. The structure of the revised article should be consistent with the basic parts of the system’s flowchart.  

2.       The “bushfire” term creates confusion for readers outside Australia. I think the term "wildfire" is more appropriate since it includes surface (grass or shrub) fires and crown fires. Note that Cruz et al. (2012) use the term “wildfire” for the Black Saturday Kilmore East fire. Accordingly, the basic types of a wildfire must be clarified in each step.

3.       Several parts of the paper have been written with much unnecessary level of detail, which could have been avoided by simply citing a relevant reference. For example, Noble’s et al. (1980) mathematical types are repeated in detail. I suggest the authors to include only the most important of them.

4.       The article presents a prototype (or a system) for fire prediction, which is based on several models of potential fire behavior. Therefore, it is not a new model as it is presented in line 9. It should be clearly stated.

5.       The manuscript describes in detail much technical information, which should be simplified in the revised form. Some of this information could be transferred to the supplementary material, or in some cases the authors could simply use a relevant reference.  

6.       There is no need to mention the content of each section at the start of the paragraph. In addition, I suggest the authors to avoid the phrase “in this study”, as it is repeated about eight times in the manuscript. 

Specific comments

Abstract

Lines 9 – 10. It is not a new fire behavior model, but a system (or prototype) for fire prediction.

Introduction

Lines 32-33. The word “and” appears twice. It should be rephrased.

Line 33. The “approaches to efficient evacuation” phrase creates confusion. It should be rephrased.

Lines 46 – 53. Please, see general comments 1 and 2. There is no need for such a separation. This paragraph may be omitted.  

Line 54. I recommend the authors to erase the “bushfire” term.

Lines 64-67. These lines could be merged with the aim of the study (start of the section).

Line 73. The 12.5 tonnes per hectare should be based on a more reliable source of information, such as a peer-reviewed paper. There is no access to the 8th reference. Fuel load is a critical parameter which actually determines fire behavior since it reveals the available fuel for combustion in the flaming front. In addition, the same reference appears twice (number 25).

Lines 74 - 103. The authors can simplify this paragraph in a general form since the structure of the prototype is described in detail in the manuscript. There is no need to repeat in detail FDIs. Instead, the authors can use the relevant references.

Line 101. Please add the following reference for WindNinja: Forthofer J (2007) Modeling wind in complex terrain for use in fire spread prediction Fort Collins, CO. Colorado State University, Ph.D.Thesis

Study area and system architecture

Line 114. It is most appropriate to use the term “wildfire” instead of “bushfire” (see general comment 2).  

Line 123-124. The term “flammability” should be based on a reference. It may involve some forest species also.

Methodology

Line 154. The authors should justify the choice of the basic fire model(s) along with the relevant references (accuracy/evaluation in real fire events). 

Line 155. One of the most important fire characteristics is the flame length, which differs from flame height (see Alexander 1982 Calculating and interpreting forest fire intensities). Please, specify the term, since only flame length incorporates the wind effect.

Lines 157 - 160. I believe that “sub-classes” is most consistent than “child-classes”.  

Line 219. I wonder, what will be the final ROS (rate of spread) when the wind direction is not in alignment with the slope, or for downslope winds.

Line 239. Table’s 4 fuel loads must be based on published values of standard fuel models or field estimates (sampling procedure). Please add the associated reference.    

Line 252. What is the scientific basis for the attributed weights presented in Table 5?

Line 305. Table 8 can be omitted. 

Line 361. I recommend the authors to transfer figure 2 to the Appendix.

Line 370. I think that the “next day” phrase can be replaced by “Afterwards” or a similar term.

Line 395. The Kappa coefficient can be used in order to validate the results (please see X. P. Rui, S. Hui, X. T. Yu, G. Y. Zhang, and B. Wu. 2018. Forest fire spread simulation algorithm based on cellular automata. Natural Hazards 91:309–319).

Line 572. It would be better to use “System” or “Prototype” instead of “model”, (please see general comment 4).

Lines 572-574. I believe that this conclusion is an obvious fact since the basic fire model has been already verified. 

Lines 610 – 616. The described limitation is typical in this kind of analyses. The RH, T, and DF affect fine fuel's moisture content, which in turn affect ROS (m/min), fireline Intensity (kW/m) and flame length (m). We faced similar problems during the development of relevant algorithms. I agree that these variables are very difficult to determine on a spatial basis. A possible solution could be based on interpolation methods or software modules, such as Solar Analyst (ArcGIS).    

Reviewer 2 Report

# Review: Dynamic Bushfire Navigation System

This paper protypes a cloud-based fire behaviour simulation model that operates on an irregular grid rather than a standard rectangular grid, in order to improve computational speed and to avoid inaccuracies that come from cellular autotomata with rectangular cells. However, I believe the authors have failed to demonstrate either the computational speed increase, or the improvement in model accuracy promised by their method. There are two major weaknesses that lead me to conclude this; firstly they do not assess their model against an existing industry fire behaviour model to compare speed and accuracy, secondly they do not assess the temporal accuracy (how well does the model predict *when* a fire will arrive) of their model at all.  I believe the paper should not be accepted in its current form - however in my comments below I have made suggestions for additional work that would be necessary properly assess the utility of the model.

## General comments

The paper presents your model as cloud-based, and draws a distinction between that and "traditional" client-server models, but it is not clear how, specifically, your moel is cloud-based.  It could be interpreted as Python code (the client) utilizing a database backend (a Postgres server); cloud computing suggests the ability to scale the processing as required, automatically, but this is not covered at all in your paper.  You do not specify, for example, how many machines your processing was run on, and you did not test and example how processing time scaled with the introduction of additional machines.  Presumably this would not scale indefinately, given a single database is providing the backend for the system.  Given the importance of rapid processing, as you stated in the paper, it would be useful to include a test/demonstration of how speed scales with the cloud architecture. In addition, a key step of the process, construction of the grids, appears to be carried out in an interactive GIS desktop (QGIS), and is not an automated part of the model (although geospatial libraries in, for example, R and Python, are capable of constructing such a grid).  Again, if managers require rapid execution of the model in order to make decisions and alert the public, manual GIS tasks should be avoided and incorporated into the model execution itself.

I have serious concerns about the processing effiency of the model; the results in table 15 state execution time on the order of dozens (or even over 100 hours) to simulate a fire of 247 km2.  This is actually an extremely slow simulation speed, compared to commonly used fire behaviour models.  Two models commonly used in Australia, Phoenix and CSIRO Spark, would be capable of simulating a similar sized fire with only minutes of processing, in the case of Phoenix using multiple worker processes on the same client machine, and in the case of Spark, by using GPU-optizied algorithms.  A run time of hours or days is unacceptably long for fire behaviour simulation, unless you are performing may replicates of the model in, for example, a Monte Carlo simulation to generate probabalistic predictions.  You will need to address this issue in the paper. 

Slower run-time may be justifiable if there is a significant increase in accuracy over existing fire behaviour models - but you have not tested any in comparison to your own. It would greatly strengthen your paper if Phoenix or Spark simulations were run on the same rectangular grid resolution as your model, so the accuracy of your model can be assessed against existing models.

In addition to the spatial accuracy/precision of fire boundaries, it is just important, when assessing fire behaviour models, to assess the *temporal* accuracy of the model - not only "did the fire burn where expected" but "did it reach various areas at the correct time".  You did not appear to assess this component of fire behaviour accuracy in this study; indeed, stopping the simulation based on total area matching the target area limits your ability to do this.  It would be more appropraite to run the simulation with a stopping time set to the same time as the actual fire was extinguished, and to incorporate intermediate isochron boundaries (which should available for this fire) to test whether the modelled fire reached them at the same time as the actual fire.  Again, given your statements regarding "directional guidance to evacuation trigger points" (line 661) - it is vital to know *when* the fire is arriving, not just *where* it's going, so your paper really needs to incorporate a temporal assessment.  It is not clear from the text but I assume the "elapse" column in table 15 is the time the fire burnt within the simulation - this varies from 11 days to 26 days, which again suggests extremely poor accuracy, which is of not much use to fire managers. 

Your testing or alternative grids to rectangular grids is useful, because of the well known weaknesses of simulating spatial processes across rectangular grids, one of the major problems being different distances between edge-neighbour centroids and corner-neighbour centroids.  You implement Voronoi and Delauney polygons as alternatives, which is potentially useful, but does introduce the additional problem of these grids being irregular.  There is a compromise that, again, I believe would greatly strengthen this paper; hexagonal grids.  Hexagonal grids have the advantages of being regular, at the same time as having equal spacing to neighbouring cells in all directions.  eg. this paper is a useful introduction. http://www.sciencedirect.com/science/article/pii/S0304380007001949 - Can you state why you believe irregular Voronoi/Delauney polygons are superior to a regular hexagonal grid?

While rectangular grids have their weaknesses, there is a significant advantage, and that is their amenability to rapid processing due to the simplicity of their simple Cartesian coordinate layout.  This enables rapid processing because they can be dealt with as a simple array with [i,j] indices by computer, without needing to rely on a spatial database to describe individual polygons.  Again, given the slow processing time of your model, I think you need to justify whether the advantage of irregular, polygonal processing units outweights the increased accuracy.  

The accuracy of the model appears relatively poor, spatially, and there is really not much difference at all between the different grids and grid sizes, in the output maps. This suggests poor accuracy was driven by something apart from the grids you have tested - input data? Modelling of spread?  I think this needs some investigation as to, for example, why you model did not simulate fire spread into the western branch of the fire at all - again, comparison with simulations by existing models such as Phoenix or Spark would help figure this out.

In terms of the fire behaviour modelling itself; you incorporate a "flammability multiplier" based on vegetation type - I have never seen this done before, could you please provide a citation for how this is justified, or other fire behaviour models doing so?  Many models now incorporate vegetation differences by including fuel loads for multiple vertical strata, incorporating structure, ladder fuels, fine vs. coarse fuels etc. and I imagine this "flammability multipler" aims to achieve a similar thing to these more physical models, but I think it needs more explanation and justification than you have given it, since the fire behaviour and rate of spread equations you use do not, naturally, have this term in them. 

## Specific Comments

* Line 11: You use the abbreviation "CV" for client-server when I believe you mean "CS".

* Line 35: You state a "high quality simulation would take to long to execute", but you really need to provide some comparison or discussion or how long other models actually *do* take to execute in order to justify this statement.  Models such as Spark or Phoenix, which are used by emergency services in real time, run a lot faster than your model seems to here. 

* Line 53: For an international audience, I think the term "wildfire" is fine. Australians know what wildfire means, non-Australians don't know what bushfire means, so it is easier to stick to the single, universally understood term. 

* Line 74: The sentance "Further, system" should read "Further, the system"

* Line 268: The descriptions you provide of Voronoi and Delaunay polygons is now clear - it would be useful to demonstrate these two tesselations, and how they are constructed, with a figure. 

* Line 350: You state the fire begins from a single ignition point; could you state whether it is possible to execute your model using multiple starting points (eg. common in lightning-lit fires), or from a starting polygon (useful for escaped burns, or for modelling an already burning fire).

* Line 410: you do not state what this "weather grid" is - I assume it is some kind of reanalysis product - what spatial/temporal resolution is it, who provided it, what numerical weather simulation model was used?  All this need to be clearly stated in the methods section

* Line 450: There is some confusion in the paper about times - on the one hand, there is how long it takes, in real time, for the model to run. On the other hand, there is the time simulated for the fire itself.  You need to make this clearer in the paper - table 15 seems to indicate "execution time" is how long it took a simulation to complete, and "elapse" is how long the fire was simulated for, but line 450 uses the term "elapsed time", as far as I can tell, to refer to the simulation execution time.  Please make sure these two time scales are referred to very carefully, because both simulation execution time, and fire burn time, are important, distinct, and easily confused. 

* Line 505; you regard the TasVeg 3.0 dataset as limited because it was created three years before the fire - in reality this is probably not an issue, as vegetation does not change rapidly in this system, and there has been no significant land clearing since that data. 

* Figure 1; this figure is not very neat; the patterbackground to the "Lake Mackenzie Road Fire" label makes this label very difficult to read, and I feel this map should probably be zoomed in closer to the fire, to show it's shape and the surrounding landscape context.
